# The environmental impact of health care for musculoskeletal conditions: A scoping review

**Bayden J. McKenzie**[1,2]*, **Romi Haas**[1,2], **Giovanni E. Ferreira**[3], **Chris G. Maher**[3], **Rachelle Buchbinder**[1,2]

**1** Department of Epidemiology and Preventive Medicine, School of Public Health and Preventive Medicine, Monash University, Melbourne, Australia, **2** Monash-Cabrini Department of Musculoskeletal Health and Clinical Epidemiology, Cabrini Health, Melbourne, Australia, **3** Institute for Musculoskeletal Health, Sydney School of Public Health, Faculty of Medicine and Health, The University of Sydney, Sydney, Australia

* bayden.mckenzie@monash.edu

## Abstract

### Background

Health care has significant environmental impact. We performed a scoping review to map what is known about the environmental impact of health care for musculoskeletal conditions.

### Methods

We included published papers of any design that measured or discussed environmental impact of health care or health support services for any musculoskeletal condition in terms of climate change or global warming (e.g., greenhouse gas emissions it produces). We searched MEDLINE and Embase from inception to 2 May 2022 using keywords for environmental health and musculoskeletal conditions, and performed keyword searches using Google and Google Scholar. Two independent reviewers screened studies. One author independently charted data, verified by a second author. A narrative synthesis was performed.

### Results

Of 12,302 publications screened and 73 identified from other searches, 122 full-text articles were assessed for eligibility, and 49 were included (published 1994 to 2022). Of 24 original research studies, 11 measured environmental impact relating to climate change in orthopaedics (n = 10), and medical aids for the knee (n = 1), one measured energy expenditure of laminar versus turbulent airflow ventilation systems in operating rooms during simulated hip replacements and 12 measured waste associated with orthopaedic surgery but did not relate waste to greenhouse gas emissions or environmental effects. Twenty-one editorials described a need to reduce environmental impact of orthopaedic surgery (n = 9), physiotherapy (n = 9), podiatry (n = 2) or occupational therapy (n = 1). Four narrative reviews discussed sustainability relating to hand surgery (n = 2), orthopaedic surgery (n = 1) and orthopaedic implants (n = 1).

**Data Availability Statement:** All relevant data are within the manuscript and its Supporting Information files.

**Funding:** BM is supported by a PhD scholarship from the Chiropractic Australia Research Foundation and a top-up scholarship from the National Health and Medical Research Council (NHMRC) Australia & New Zealand Low Back Pain Research Network Centre of Research Excellence (ANZBACK CRE) (1171459). GF is supported by an NHMRC Emerging Leadership Fellowship (APP2009808). CM and RB are supported by NHMRC Leadership Fellowships. The funders had no role in study design, data collection and analysis, decision to publish, or preparation of the manuscript. There was no additional external funding received for this study. Funder websites: https://chiropracticaustralia.org.au/research-foundation/, https://anzback.org/, https://www.nhmrc.gov.au/.

**Competing interests:** The authors have declared that no competing interests exist.

## Conclusion

Despite an established link between health care and greenhouse gas emissions we found limited empirical data estimating the impact of musculoskeletal health care on the environment. These data are needed to determine whether actions to lower the carbon footprint of musculoskeletal health care should be a priority and to identify those aspects of care that should be prioritised.

## Introduction

Climate change is an existential crisis [1]. There is a need to understand the key contributors to climate change to minimise their impacts. Health care results in significant direct and indirect greenhouse gas emissions, commonly termed the 'carbon footprint'. It is responsible for between one to five percent of the total global environmental impacts [2], although the proportion of overall greenhouse emissions due to health care is greater in some countries such as the United States (8.5%) [3], and Australia (7%) [4]. The UK, whose health care greenhouse emissions is responsible for approximately 4% of the UK's footprint, is leading the world in striving for carbon neutral health care by 2040 [5].

The largest contributors to the carbon footprint of health care are generated as part of hospital stays, surgery, pharmaceutical manufacturing and imaging [4, 6]. Recent carbon footprint estimates suggest the majority of health care related greenhouse emissions are produced from energy use and the health care supply chain such as manufacturing medical equipment and materials, transport, agriculture and waste disposal [7]. Yet awareness of the carbon footprint generated by different aspects of health care is not yet well appreciated among many health care providers or the general public, delaying efforts to identify and reduce it [8].

Approximately one third of health care is estimated to be of low value or 'wasted' [9–11]. For example, there is a large body of evidence attesting to widespread low-value health care practices for common musculoskeletal conditions such as osteoarthritis [12], low back pain [13], hip and knee pain [12], shoulder pain [14–16] and sports injuries [17]. Directing efforts towards eliminating these aspects of care would have the dual benefit of reducing harms associated with unnecessary care, and avoiding their harmful effects on the environment.

While the environmental impact of health care in some fields of medicine has been investigated, including treatment of patients with septic shock in intensive care [18], cataract surgery [19] and geriatric medicine [20], there is a paucity of evidence outlining environmental impacts of other types of care. The aim of this scoping review was to map what is known about the environmental impact of health care for musculoskeletal conditions.

## Methods

We reported this scoping review in accordance with the recommendations of the Preferred Reporting Items for Systematic reviews and Meta-Analyses extension for Scoping Reviews (PRISMA-ScR; [21], see S1 Data).

### Selection criteria and study selection

We included published papers that measured or explicitly discussed the environmental impact of health care or health support services for any musculoskeletal condition. This could include the impacts of the care (e.g. imaging, hospital visits, surgery, prescription medication) on

indices of climate change or global warming such as the amount and type of greenhouse gas it produces [22]. All publication designs were eligible for inclusion, including original research, reviews, or commentaries. We did not impose any date or language restrictions.

## Search strategy

We searched electronic databases of MEDLINE and Embase (via Ovid) from inception to 2 May 2022. Our search strategy consisted of combining two concepts: environmental health and musculoskeletal conditions. Internet searches were also performed using Google and Google Scholar between 2 May and 12 May 2022 within the Google Chrome browser. The internet search engines were chosen to ensure a wide range of publications across multiple musculoskeletal disciplines, and combined environmental keywords 'life cycle assessment', 'sustainability', 'environmental sustainability', 'environmental impact' and 'carbon footprint' with terms of 'hand', 'wrist', 'elbow', 'shoulder', 'foot', 'ankle', 'knee', 'hip', 'spine' and 'spinal'. We also used various combinations of the following keywords: 'surgery', 'surgical', 'surgical implant', 'orthopaedic surgery', 'joint arthroplasty', 'joint arthroscopy', 'joint replacement', 'telemedicine' and 'telehealth'. We considered the first 50 Google and Google Scholar results from each set of keywords. The full search strategy is presented in see S2 Data. We also hand searched reference lists of included publications.

All records generated from electronic databases were exported to Covidence (Veritas Health Innovation, Melbourne, Australia) for duplicate removal and screening [23]. Two authors (BM and either RH, GF, CM or RB) independently assessed each title and abstract and then independently screened the full texts of potentially eligible publications to identify those eligible for inclusion. Google and Google Scholar records were independently assessed by one author (BM). Potentially eligible publications were downloaded as full texts and screened by two authors (BM and either RH, GF, CM or RB). Publications not written in English were translated with Google Translate [24]. Conflicts were resolved through discussion. Publications relating to the same primary publication were considered together and counted only once.

## Data charting and analysis

For each original research publication, one author (BM) independently charted author/s, year, country, setting, timing of study, study design, topic, aim/s and methods, results and conclusion. Data from editorial publications and narrative reviews were independently charted by one author (BM) for author/s, year, country of author/s, topic, focus and conclusions. Another author (RH, GF or RB) independently verified all data extraction. A narrative synthesis of the papers is presented.

## Results

Of 12,302 publications retrieved and screened from electronic databases and 73 publications that were identified and screened using Google, Google Scholar and hand searches of citations, 122 full-text articles were assessed for eligibility and 48 were excluded (Fig 1). Reasons for exclusion were wrong topic (n = 25) [25–49], wrong setting (n = 3) [50–52], wrong population (n = 1) [53], unclear population (n = 4) [54–57], and duplicates (n = 15) (see S1 Table and S2 Data for more detail). Nine conference abstract publications were classified as awaiting assessment [58–66] (see S2 Table). Sixteen reports of included publications were collated with an associated primary report and counted as a single unit to prevent duplication of the same record [67–82] (see Tables 1 and S3). Forty-nine primary publications were included in this review [83–131].

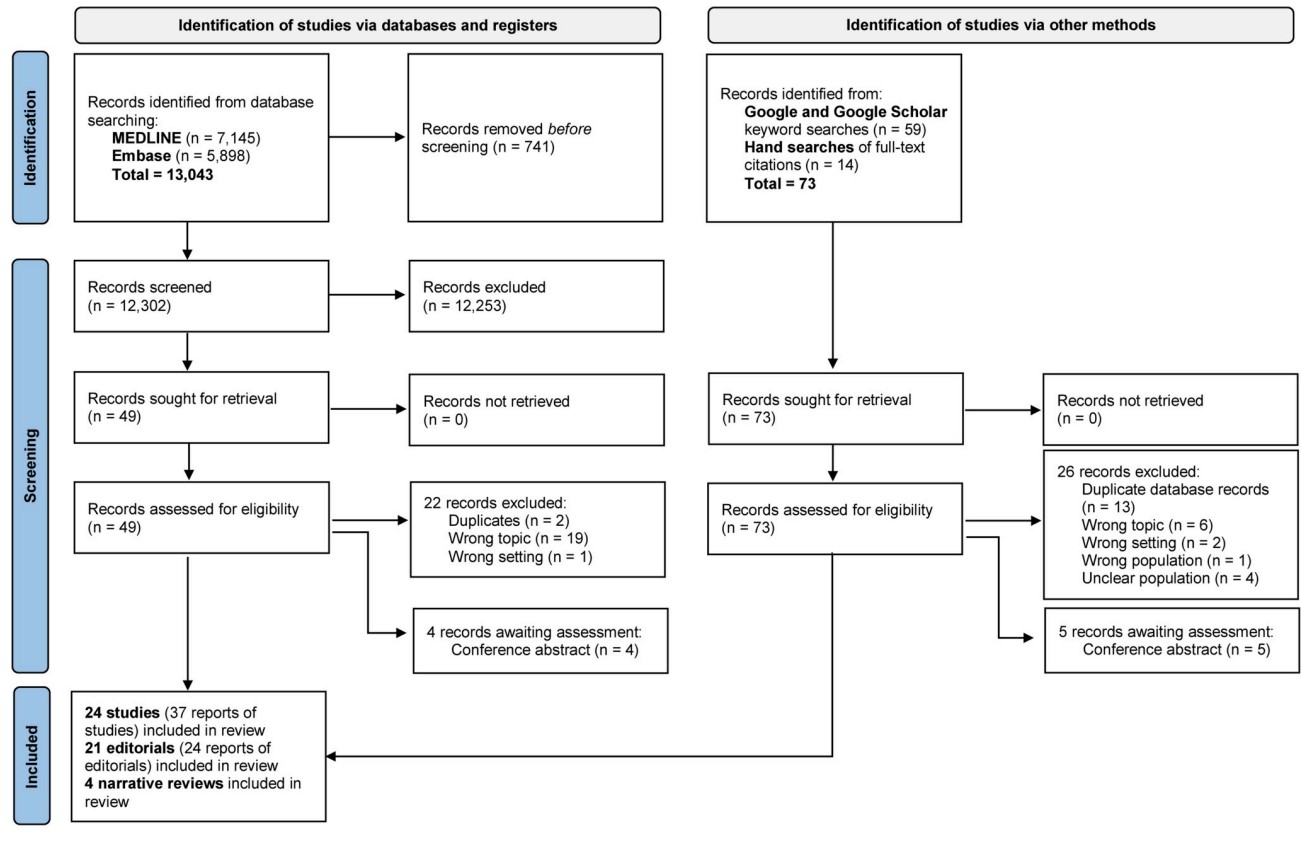

**Fig 1. PRISMA flowchart.**

## Overview of included publications

The characteristics and findings of included publications are presented in Table 1 for original research publications and as supplementary materials for included editorials and reviews (see S3 and S4 Tables). Included papers were published from 1994 to 2022, with most published since 2019 (n = 36, 73%).

There were 24 original research papers, nine from the United States [83–91], three from Canada [92–94], two each from Ireland [95, 96], Sweden [97, 98] and the United Kingdom [99, 100], and one each from Australia [101], Denmark [102], Germany [103], Italy [104], Serbia [105], and South Korea [106].

There were 21 editorials with authors from 13 countries; Australia [107–111], Brazil [109], Canada [112], Germany [109], Greece [109], India [113, 114], New Zealand [109, 115], Norway [109, 111, 114, 116], Pakistan [109], Sweden [109, 117, 118], Switzerland [109, 119], United Kingdom [100, 109, 117, 120–122] and United States [109, 116, 123–127]. There were four narrative reviews, two from the United States [128, 129] and one each from India [130] and the United Kingdom [131]. Thirty-six (73%) included publications were related to orthopaedic surgery [83–104, 106, 117, 120–131].

**Original research studies.** Eleven (46%) of the 24 research studies used a life cycle assessment (LCA) or carbon footprinting methodology to measure the environmental impact of health care or health support services in the fields of orthopaedics (n = 9) [83–86, 91, 95, 97, 101, 103, 104], and medical aid manufacturing (n = 1) [105].

**Table 1. Characteristics and findings of original publications.**

| Author (year) | Country, setting and time of study | Study design | Topic | Aim/s and methods | Results | Conclusion |
|---|---|---|---|---|---|---|
| **Baxter *et al.* 2021** [83] | United States<br><br>**Setting:** 19 institutions<br><br>**Timing:** February 2020 | Survey and life cycle assessment | Orthopaedic surgery | To investigate how variation in use of disposable surgical supplies contributes to environmental and financial burden. Surgeons completed survey relating to (i) carpal tunnel release, (ii) open reduction and internal fixation of distal radius fracture, or (iii) primary flexor tendon repair. | **Number of participants**: 35 (54 invited)<br><br>Carbon emissions per hand surgery procedure ranged from 7.8 to 28.8 kg per person, and were 10.9 kg greater for high-use versus lean-use surgeons. | There are opportunities to reduce the carbon footprint of hand surgery.<br><br>Surgeons support sustainable practice but underestimate the environmental impact of surgery. |
| **Bravo *et al.* 2022** [84] | United States<br><br>**Setting:** Surgical centre affiliated with a large academic centre<br><br>**Timing:** February 2018 to July 2018 | Carbon footprint study (using hospital purchasing records and an environmentally extended input-output (EEIO) life cycle assessment) | Orthopaedic surgery (hand surgery waste) | To identify sources of unnecessary waste and decrease costs of care by analysing quantity, cost and greenhouse gas emissions of opened and unused disposable surgical supplies. | **Total surgeries:** Convenience sample of 85 cases of hand surgery (endoscopic carpal tunnel release n = 45; tendon transfers, tenolysis, tendon sheath incisions n = 30; open reductions of distal radius and internal fixations n = 7; and carpal bone and phalangeal fractures n = 3)<br>**Mean (SD) number of wasted surgical items from 51-item custom surgical pack:** 11.5 (3.6)<br>**Total number of wasted items:** 981<br>**Total weight of wasted items:** 441 kg of carbon dioxide equivalent ($CO_2$e) emissions | Environmental impact and costs of hand surgery can be reduced by creating awareness of unnecessary waste.<br><br>Approaches to reduce waste include (i) reducing number of items available in operating room, (ii) correctly sorting waste for disposal and recycling, (iii) optimising supply of surgical items and (iv) incorporating environmentally conscious initiatives. |
| **Cappucci *et al.* 2020** [104]<br><br>Record related to Cappucci *et al.* [67] | Italy<br><br>**Setting:** Engineering laboratory<br><br>**Timing:** not reported | Life cycle assessment | Orthopaedic surgery (titanium hip prosthesis manufacturing) | To assess environmental impacts of titanium (Ti-6AI-4V) alloy-based femoral stems produced with additive manufacturing (AM) over their entire life cycle, to (i) identify environmental hotspots, and (ii) compare any benefits to traditional manufacturing processes | Based on a life cycle impact assessment (LCIA) for the manufacture of 1 femoral stem (hip prosthesis) with gas atomisation (GA) powder, 'global warming' impact at mid-point level was:<br>**Production phase:** 38.8 kg $CO_2$e (69.3% of total environmental burden)<br>**Use phase:** 17.6 kg $CO_2$e (30.6% of total environmental burden)<br>**End of life phase:** 0.0175 kg $CO_2$e (0.03% of environmental burden)<br>**Total:** 56.4 kg $CO_2$e | The additive manufacturing process was more sustainable for titanium femoral stem manufacturing due to recovery of loose titanium power at the end of the process that can be reused. |

*(Continued)*

**Table 1.** (*Continued*)

| Author (year) | Country, setting and time of study | Study design | Topic | Aim/s and methods | Results | Conclusion |
|---|---|---|---|---|---|---|
| **Leiden *et al.* 2020** [103] | Germany<br><br>**Setting:** Engineering laboratory<br><br>**Timing:** not reported | Life cycle assessment | Orthopaedic surgery (disposable versus reusable surgical instruments for lumbar spine fusion) | To investigate environmental impact of a disposable and reusable instrument set. | **Key findings**[1]<br>• The environmental impact of disposable instrument sets was ~45 to 85% less than reusable instrument sets in all impact categories of data collection (cumulative energy demand (CED), abiotic depletion potential (ADP), global warming potential (GWP), acidification potential (AP) and particulate matter (PM)) and the single score indicator (ReCiPe Endpoint)<br>• Steam sterilisation for reusable instruments was the greatest contributor of greenhouse gas emissions due to energy use<br>• The production phase of the disposable instrument set was the greatest contributor of greenhouse gas emissions; these were consistently higher compared to the reusable set<br>• The environmental impact of transport and disposal of waste processes was minimal across the life cycle of both surgical instrument types | Environmental impact of the disposable surgical instrument set was lower than the reusable set, mostly related to the high environmental impact of the steam sterilisation process. |
| **Lyons *et al.* 2021** [95] | Ireland<br><br>**Setting:** Engineering laboratory<br><br>**Timing:** not reported | Life cycle assessment | Orthopaedic surgery (titanium knee implant manufacturing) | To compare environmental impact (primary energy consumption (PEC) and $CO_2$ emissions) of manufacturing titanium Ti-6Al-4V femoral components used in typical knee implants via additive (using electron beam melting (EBM) methods) versus conventional (using milling methods) manufacturing. | **Carbon dioxide ($CO_2$) emissions**<br>**Additive manufacturing**<br>• Production of Ti-6AI-4V powder: 11.47 kg per part<br>• Electron beam melting: 3 kg per part<br>• Post-process milling: 0.41 kg per part<br>• Post process grinding: 0.06 kg per part<br>• **Total $CO_2$ emissions: 14.94 kg per part**<br><br>**Conventional manufacturing**<br>• Production of Ti-6AI-4V workpiece: 45.24 kg per part<br>• Roughing: 0.68 kg per part<br>• Finishing: 0.89 kg per part<br>• Post process grinding: 0.11 kg per part<br>• **Total $CO_2$ emissions: 46.92 kg per part** | Manufacture of a titanium knee implant using additive methods was more environmentally sustainable largely due to greater efficiencies and less waste compared with conventional methods. |

(*Continued*)

**Table 1.** (Continued)

| Author (year) | Country, setting and time of study | Study design | Topic | Aim/s and methods | Results | Conclusion |
|---|---|---|---|---|---|---|
| | | | | | **Total PEC**<br>**Additive manufacture:** 143.22 MJ per part<br>**Conventional manufacture:** 314.52 MJ per part | |
| **McGain et al. 2021 [101]** Records commenting on McGain et al. [68–74] | Australia<br><br>**Setting:** Hospital<br><br>**Timing:** Between 9 January 2019 and 10 June 2019 | Life cycle assessment | Orthopaedic surgery (general, regional and combined anaesthesia) | To quantify the carbon dioxide equivalent emissions of three anaesthetic approaches for total knee replacement surgery. | **Total surgeries:** Convenience sample of 30 total knee replacements (general anaesthesia n = 10, spinal anaesthesia n = 10, combined general and spinal anaesthesia n = 10)[2]<br><br>**Anaesthesia and mean $CO_2e$ emissions per patient**[3]<br>*General anaesthesia group*: sevoflurane gas (n = 8), total intravenous anaesthesia (n = 1), sevoflurane/total intravenous anaesthesia (n = 1): 14.9 kg $CO_2e$ (95% CI, 9.7 to 22.5)<br>*Spinal anaesthesia group*: propofol infusion (n = 10): 16.9 kg $CO_2e$ (95% CI, 13.2 to 20.5)<br>*Combined general and spinal anaesthesia group*: sevoflurane gas + propofol (n = 6), total intravenous anaesthesia + propofol (n = 3): 18.5 kg $CO_2e$ (95% CI, 12.5 to 27.3) | The average carbon footprint of anaesthesia for a knee replacement was similar for general, spinal and combination approaches when sevoflurane was the inhaled anaesthetic gas used for general and combination approaches with an average low fresh gas flow.<br>The carbon footprint of knee replacement surgery can be reduced by using low-flow anaesthetic gas and/or local anaesthesia, reducing single-use plastics and oxygen flows during surgery, and collaborating with engineers to improve energy efficiency. |
| **Vukelic et al. 2017 [105]** | Serbia<br><br>**Setting:** Engineering laboratory<br><br>**Timing:** 2014 to 2015 | Life cycle assessment (cradle to gate; case study) | Medical aid (knee support brace) | To develop a multi-criteria decision-making model for optimal product selection of 3 types of knee support (elastic, crossed and hinged) using life cycle assessment (LCA) and multi-criteria decision making (MCDM) approaches. | Based on LCA results, elastic knee support production had the lowest environmental impact, followed by the crossed knee support. Polyester was identified as the highest contributor to the environmental impact for each knee support.<br><br>The MCDM-LCA model output ranked the elastic knee support as the best, followed by the hinged knee support and then the crossed knee support.<br><br>Results differences between LCA and MCDM-LCA approaches were due to the significant weighting of economic and technical criteria for the MCDM-LCA model. | LCA and MCDM approaches can identify knee supports with the lowest environmental impact and can be used to optimise 'eco-design' of new knee support products. |

*(Continued)*

**Table 1.** (Continued)

| Author (year) | Country, setting and time of study | Study design | Topic | Aim/s and methods | Results | Conclusion |
|---|---|---|---|---|---|---|
| **Wang et al. 2021 [91]** | United States<br><br>**Setting:** Pre-operative evaluation centres<br><br>**Timing:** two-month pre-intervention period (Sept-Oct 2015) and two-month post-intervention period (Sept-Oct 2016) | Life cycle assessment (retrospective cross-sectional analysis) | Telehealth (spinal surgery) | To determine the greenhouse gas emissions associated with pre- and post-implementation of a novel telehealth preoperative evaluation centre (PEC) in patients undergoing elective spine surgery. | **Number of included patient records:** 298[4]<br>**Study intervention:** New PEC model including telehealth (phone) evaluations and standardised preoperative testing guidelines versus traditional preoperative care where a surgeon decides which patients require in-person PEC evaluation.<br>**Mean (SD) pre-intervention kg $CO_2$e per patient (n = 144):**<br>Testing (e.g., blood tests & imaging): 15.65 (0.63)<br>Physician in-person evaluation: 11.77 (0)<br>PEC: 18.70 (1.74)<br>Telehealth: 1.16 (0.18)<br>Vehicular travel: 37.22 (3.01)<br>Total: 84.52 (3.31)<br><br>**Mean (SD) post-intervention kg $CO_2$e and t-test of difference to pre-intervention (n = 154)**<br>Testing: 12.83 (0.21), p<0.001<br>Physician in-person evaluation: 11.77 (0), p>0.05<br>PEC: 3.99 (0.84), p<0.001<br>Telehealth: 8.82 (0.38), p<0.001<br>Vehicular travel: 39.01 (3.15), p = 0.56<br>Total: 76.43 (3.54), p = 0.019 | Implementing a telehealth preoperative evaluation process with standardised preoperative testing guidelines led to reduced carbon emissions. |
| **Zhang et al. 2022 [85]** | United States<br><br>**Setting:** large multicentre, urban health system in a single US metropolitan region<br><br>**Timing:** Data were retrospectively identified from 2020 | Life cycle assessment | Orthopaedic surgery (carpal tunnel release) | i) To quantify the carbon footprint of carpal tunnel surgery and its principal driving components. ii) To compare the carbon footprint of open versus endoscopic carpal tunnel release. | **Total surgeries:** 28 (14 open, 14 endoscopic)<br>**Mean (SD) carbon footprint (in kg $CO_2$e) for open versus endoscopic:**<br>Central processing[5]: 40.7 (0) vs 81.4 (0)<br>Facility[6]: 18.5 (5.5) vs 24.6 (6.3)<br>Solid waste[7]: 0.4 (0.2) vs 0.5 (0.2)<br>Total carbon footprint: 59.6 (5.7) vs 106.5 (6.4), P<0.05<br>Average duration of time in operating room significantly shorter for open (38 vs 49 minutes, P<0.05). | Endoscopic carpal tunnel release was associated with a larger carbon footprint across all categories. |

*(Continued)*

**Table 1.** (Continued)

| Author (year) | Country, setting and time of study | Study design | Topic | Aim/s and methods | Results | Conclusion |
|---|---|---|---|---|---|---|
| **Holmner *et al.* 2014 [97]** <br><br> Records related to Holmner *et al.* [75–77] | Sweden <br><br> **Setting:** Hospital <br><br> **Timing:** January to December 2012 | Simplified, streamlined life cycle inventory | Telemedicine (hand surgery) | To evaluate potential of telemedicine to reduce carbon emissions for hand rehabilitation consultations following a range of hand surgeries. | **Number of consultations:** 238 (81 from patient's home using PC or tablet, 157 from nearest primary health centre using videoconferencing equipment). **Accumulated life cycle carbon costs of car travel for face-to-face visits (n = 238):** 21,400 kg $CO_2e$[10] or 42,472 kg $CO_2e$[11]. **Accumulated life cycle carbon costs of 1-hour telemedicine consultations (n = 238):** 602 kg $CO_2e$ (range: 183 to 1364). | Telemedicine can significantly reduce carbon emissions vs face to face care for hand surgery rehabilitation. |
| **Wang *et al.* 2022 [86]** <br><br> Record related to Wang *et al.* [78] | United States <br><br> **Setting:** Academic centre <br><br> **Timing:** not reported | Carbon footprint study (retrospective medical chart review) | Orthopaedic surgery (spinal fusion) | i) To compare carbon emissions of general vs spinal anaesthesia for single-level spinal fusion | **Total surgeries:** 100 by a single surgeon (50 general and 50 spinal anaesthesia) **Median total carbon footprint, grams $CO_2e$** *Spinal anaesthesia*[12]: 70 *General anaesthesia (n = 50)*[12]: 4,725 *Sevoflurane only*[12]: 4,802 *Desflurane only (n = 6)*[12,13]: 154,008 | Spinal anaesthesia had significantly less environmental impact than general anaesthesia with the impact being greater for desflurane than sevoflurane. |
| **Marsault *et al.* 2021 [102]** | Denmark <br><br> **Setting:** Hospital <br><br> **Timing:** 31 October 2014 to 30 January 2015 (on Fridays) | Simulation study | Orthopaedic surgery (airflow and energy consumption in operating rooms) | To determine the energy consumption, bacteria and particle counts of large, high-volume, laminar airflow (LAF) and turbulent airflow (TAF) ventilation systems at 100% and 50% fresh air influx during standardised simulated total hip arthroplasty. | **Total surgeries:** 32 standardised simulated total hip arthroplasties (LAF 100% n = 8, LAF 50% n = 8, TAF 100% n = 8, TAF 50% n = 8) **Energy consumption (kWh) with 100% fresh air ventilation:** LAF: 1.85 kWh (1.66 to 2.03) TAF: 1.54 kWh (1.53 to 1.83) **Energy consumption (kWh) with 50% fresh air ventilation:** LAF: 1.12 kWh (0.95 to 1.31) TAF: 0.75 kWh (0.73 to 0.87) | Lowering fresh air influx of laminar air flow (LAF) by 50% significantly lowered energy consumption but did not adversely affect the bacterial or particle counts. |

*(Continued)*

**Table 1.** (Continued)

| Author (year) | Country, setting and time of study | Study design | Topic | Aim/s and methods | Results | Conclusion |
|---|---|---|---|---|---|---|
| **Albert & Rothkopf 2015** [87] | United States<br><br>**Setting:** Hospital (University of Massachusetts)<br><br>**Timing:** January 2012 to April 2013 | Before-after study | Orthopaedic (hand) and plastic surgery | To propose a method of decreasing cost through judicious selection of instruments and supplies, and initiation of recycling in hand and plastic surgery. A redesigned 'operating set' was implemented after removing items that were routinely opened and wasted. | **Total surgeries:** not reported<br><br>**Surgery types**[14]**:** carpal tunnel release, ganglion cyst excision, A1 pulley release for trigger finger, Dupuytren's contracture excision, tendon repair, fracture open reduction and internal fixation, fracture closed reduction, and percutaneous pinning.<br><br>**Mean recycling rates for hand and plastic surgeries over 9 months (from April 2013):**<br>• *Hahnemann campus* 4.28 tonnes/month (recycling rate 51%)[15]<br>• *University campus* 37 tonnes/month (recycling rate 29%)<br>• *Memorial campus* 8.84 tonnes/month (recycling rate 20%) | Significant environmental benefit (and financial savings) can result by altering surgical disposable packs and instrument sets and by implementation of recycling. |
| **de SA *et al.* 2016** [92]<br><br>Record related to de SA *et al.* [79] | Canada<br><br>**Setting:** Hospital<br><br>**Timing:** March 2015 to April 2015 | Hospital waste audit | Orthopaedic surgery (surgical waste and recycling) | To identify potential waste reduction practices. | **Total surgeries:** 5 hip arthroscopies for femoroacetabular impingement[16]<br><br>**Mean waste weight per surgery:** 9.48 kg (excluding laundered linens that are cleaned and reused)<br>• 1.28 kg (13.5%) normal solid waste<br>• 4.34 kg (45.7%) biohazard waste<br>• 2.34 kg (24.7%) sterile wrap (recyclable)<br>• 1.28 kg (13.5%) recyclable plastic<br>• 0.24 kg (2.6%) sharps<br><br>**Data extrapolation:** Based on estimates of 500 hip arthroscopies performed for femoroacetabular impingement in Ontario, Canada, approximately 4,700 kg of waste is produced each year. This equates to approximately 18,800 kg of waste produced from approximately 2,000 of these procedures performed in Canada every year. | Femoracetabular impingement procedures produce considerable biohazard waste that could be reduced by recycling programs, adherence to proper waste segregation and emphasising 'green outcomes.' |

*(Continued)*

**Table 1.** (Continued)

| Author (year) | Country, setting and time of study | Study design | Topic | Aim/s and methods | Results | Conclusion |
|---|---|---|---|---|---|---|
| **Hennessy *et al.* 2021** [96] | Ireland<br><br>**Setting:** Hospital<br><br>**Timing:** July 2018 to July 2019 | Hospital waste audit | Orthopaedic surgery (surgical waste and recycling) | To assess the burden of waste associated with implant packaging in Galway University Hospital operating theatres. | **Total surgeries**[17]: 1 open reduction internal fixation for malleolus ankle fracture.<br>**Surgical waste weight**[18]: 211 g<br>• Cardboard (not recyclable): 144 g (68%)<br>• Hard plastic (recyclable): 42 g (20%)<br>• Soft plastic (not recyclable): 25 g (12%)<br><br>**Data extrapolation:** Based on one standard procedure, 209 procedures produce over 44 kg of surgical waste at the study hospital in one year. | Orthopaedic implants contribute a significant amount of operative waste that could be reduced by reducing volume and layers of packaging for surgical materials and using kits which can be re-sterilised between procedures. |
| **Kooner *et al.* 2020** [93] | Canada<br><br>**Setting:** 1 adult and 1 paediatric tertiary care hospital<br><br>**Timing:** November 2017 (1-month period) | Hospital waste audit | Orthopaedic surgery (surgical waste and recycling) | To determine the amount of waste produced in the preoperative and intraoperative periods for several orthopaedic subspecialties and to assess how much surgical waste was recycled. | **Total surgeries:** 55; joint replacement (n = 14), sports (n = 10), trauma (n = 10), upper extremity (n = 12), foot and ankle (n = 4), paediatrics (n = 5).<br>**Mean waste weight per surgery:** 6.2 kg (95% CI 3.75 to 8.30)<br>• 27% recyclable, 70% non-recyclable, 3% biological<br>• 71% waste in the intraoperative period, of which 8% was recyclable<br>• 29% waste in the preoperative period, of which 74% was recyclable<br>**Mean waste weight per joint replacement surgery:** 8.8kg (95% CI 8.48 to 9.07)<br>• 34% recyclable<br>• 86% recyclable waste in the preoperative period and 14% in the intraoperative period<br><br>**Data extrapolation:** Based on an estimated 7 million orthopaedic procedures in the US per year, 11,564,000 kg of landfill waste could be diverted for recycling each year (>2 tonne waste from total joint replacement surgeries). | Orthopaedic surgery is a substantial source of waste production in the hospital system. Nearly 3/4 of all waste in the preoperative period can be effectively recycled.<br><br>Joint replacement surgery is one of the largest waste producers, but it also has the highest potential for recycling of materials. |

(*Continued*)

**Table 1.** (*Continued*)

| Author (year) | Country, setting and time of study | Study design | Topic | Aim/s and methods | Results | Conclusion |
|---|---|---|---|---|---|---|
| **Lee & Mears 2012 [88]**<br><br>Record commenting on Lee & Mears [82] | United States<br><br>**Setting:** Hospital<br><br>**Timing:** March to April 2011 (2 months) | Hospital waste audit | Total hip and knee joint replacements (surgical waste) | To determine which types of waste produced during hip and knee replacement surgeries can be recycled | **Total surgeries:** 20 consecutive primary total hip (n = 10) and knee (n = 10) replacements **Mean (range) waste per hip replacement, kg[19]:** 13.6 (12.3 to 14.8) Non-recyclable (contaminated) waste[19,20]: 9.5 (8.4 to 10.4) Uncontaminated waste[19,21]: 4.1 (3.5 to 5.1), includes 22.8% potentially recyclable paper and plastic material **Mean (range) waste per knee replacement, kg[19]:** 15.1 (14.0 to 16.0) Non-recyclable (contaminated) waste[19,20]: 10.6 (9.6 to 11.5) Uncontaminated waste[19,21]: 4.5 (3.3 to 5.3), includes 22.0% potentially recyclable paper and plastic material | Thirty percent of operating room waste produced during hip and knee joint replacements is clean and uncontaminated, of which one-fifth can be recycled. |
| **McKendrick et al. 2017 [99]** | United Kingdom<br><br>**Setting:** Hospital<br><br>**Timing:** Not reported | Hospital waste audit | Orthopaedic surgery (surgical waste and recycling) | (i) To measure the volume and weight of paper and cardboard which could be recycled within an operating theatre environment. (ii) To calculate the potential cost and environmental savings which might result from recycling paper and cardboard. | **Total surgeries:** 20 consecutive orthopaedic surgeries; major (n = 12), minor (n = 8). **Surgery types:** not reported. **Total waste weight for 20 surgeries by waste type, kg (%):** • **Overall total waste: 218 (100)** • Clinical waste: 144 (66) • General (landfill) waste: 20 (9) • Recyclable paper: 40 (18) • Recyclable cardboard: 14 (6)<br><br>**Data extrapolation:** Based on an estimated 23 tonnes of recyclable paper and cardboard produced at the study hospital in 2013–14, $CO_2$ emissions could be reduced by 11 tonnes annually. | Recycling paper and cardboard waste from the anaesthetic room and theatre preparation room has significant environmental and financial benefits. |
| **Rammelkamp et al. 2021 [89]** | United States<br><br>**Setting:** Medical centre **Timing:** September 2019 (5 days, 9am to 5pm) and December 2019 (5 days, 9am to 5pm) | Hospital waste audit | Surgery (surgical waste) | To determine the amount of waste from musculoskeletal surgeries from two five-day audits. | **Total musculoskeletal surgeries[22]:** 50; total knee replacement (n = 14), laminectomy (n = 6), total shoulder replacement (n = 6), amputation (n = 6), total hip replacement (n = 3), carpal tunnel release (n = 2), gastrocnemius repair (n = 2), | Most surgical waste was non-recyclable (on average 85% general and 2% biohazardous).<br><br>Conducting hospital waste audits may drive a systems approach to reduce waste, and lead to environmentally sustainable health care practices. |

(*Continued*)

**Table 1.** (Continued)

| Author (year) | Country, setting and time of study | Study design | Topic | Aim/s and methods | Results | Conclusion |
|---|---|---|---|---|---|---|
| | | | | | Dupuytren's contracture excision (n = 2), ACL repair (n = 1), foraminotomy (n = 1), fasciotomy (n = 1), ankle ligament repair (n = 1), ankle open reduction internal fixation (n = 1), volar wrist repair (n = 1), arthrodesis (n = 1), rotator cuff repair (n = 1) and arthroscopic clavicle repair (n = 1). **Mean waste weight per musculoskeletal surgery (n = 50), kg[23]:** • Total waste: 11.8 (1.1 to 24.3) • General waste: 9.9 (1.1 to 16.7) • Recyclable waste: 1.5 (0.0 to 3.8) • Biohazard waste: 0.2 (0.0 to 7.1) • Blue wrap: 0.7 (0.0 to 5.1) **Mean waste weight per joint replacement surgery (n = 23), kg:** • Total waste: 15.0 (7.4 to 24.3) • General waste: 12.2 (6.9 to 15.1) • Recyclable waste: 1.0 (0.0 to 3.8) • Biohazard waste: 0.3 (0.0 to 7.1) • Blue wrap: 1.0 (0.0 to 2.2) | |
| **Sand Lindskog et al. 2019 [98]** | Sweden<br><br>**Setting:** Surgery departments at three hospitals<br>**Timing:** 2013–2014 | Survey and hospital waste audit | Orthopaedic surgery (waste reduction to reduce climate impact) | To reduce the environmental impact of health and medical care in Sweden based on a European Union waste policy framework that includes waste prevention, waste management and improving resource efficiency such as packaging and procurement of surgical materials. | **Total surgeries:** not reported<br>**Surgery type:** total hip joint replacement (cemented)<br>**Mean waste weight per surgery, kg:** 5.7 (5 to 6.6)<br>Based on the variation in techniques between the 3 hospitals, the authors estimated waste could be reduced to:<br>• 4.5 kg/surgery if all operating departments used the lowest product consumption (most slim material)<br>• 3.9 kg/surgery if all departments changed from disposable to reusable materials | The study led to the introduction of customised operating kits for total hip replacement surgery that are adapted to the needs of different hospitals and types of surgery in order to reduce the amount of sterile packaging. However, the rationale for these customised operating kits and the calculation of how much waste it would reduce is unclear. |

(*Continued*)

**Table 1.** (*Continued*)

| Author (year) | Country, setting and time of study | Study design | Topic | Aim/s and methods | Results | Conclusion |
|---|---|---|---|---|---|---|
| **Shinn *et al.* 2017** [106] | South Korea<br><br>**Setting:** Hospital<br>**Timing:** June 2015 | Hospital waste audit | Orthopaedic surgery (surgical waste) | To identify the amount and type of waste produced by operating rooms in order to reduce the hospital–regulated medical waste so as to achieve environmentally friendly waste management in the operating room. | **Total surgeries:** 5 total joint replacements[24]; knee (n = 4) and hip (n = 1)<br>**Mean waste weight and estimated volume per surgery:** 16.9 kg, 240.4L<br>• 3.3 kg (19.4%), 133.6L (55.6%) non regulated medical waste<br>• 12.6 kg (74.4%), 90.6L (37.7%) regulated medical waste<br>• 1.0 kg (6.2%), 16.2L (6.7%) blue wrap<br>**Data extrapolation:** Based on 105 total knee replacement surgeries and 97 total hip replacement surgeries conducted at the study hospital in 2014, 872.6 kg of regulated medical waste can be reduced by waste segregation. | "It is possible to reduce the amount of hospital regulated medical waste through the segregation of waste in the operating room. This gives clinicians the opportunity to deliberately plan a way to balance the importance of patient care with consideration for the impact on the environment." |
| **Southorn *et al.* 2013** [100] | United Kingdom<br><br>**Setting:** Two hospitals<br><br>**Timing:** 2-week period | Hospital waste audit | Orthopaedic surgery (surgical waste) | To examine the effect of separating and recycling surgical waste to reduce incinerated waste (implied). | **Total surgeries or invasive procedures:** 44; total hip replacement (n = 18), total knee replacement (n = 14) and facet joint injections (n = 12).<br>**Mean waste weight per total hip replacement, kg (SD):** 12.1 (0.25), which includes 5.8 (0.17) of domestic waste<br>**Mean waste weight per total knee replacement, kg (SD):** 11.6 (0.18), which includes 5.3 (0.18) of domestic waste<br>**Mean waste weight per facet joint injection kg (SD):** 1.8 (0.17), which includes 0.8kg (0.20) of domestic waste<br>Domestic waste was predominantly comprised of recyclable materials[25]<br>**Data extrapolation:** Based on 180,000 joint replacements performed in the UK each year, the carbon footprint of joint replacements would be reduced by 75% (6.3 million kg of carbon dioxide) if waste was separated and recycled rather than being incinerated. | Changing clinical practice to recycle domestic operating theatre waste can have a positive impact on the environment and significantly reduce costs. |

(*Continued*)

**Table 1.** (Continued)

| Author (year) | Country, setting and time of study | Study design | Topic | Aim/s and methods | Results | Conclusion |
|---|---|---|---|---|---|---|
| **Stall et al. 2013** [94] | Canada<br><br>**Setting:** Hospital<br><br>**Timing:** February 2010 (1-month period) | Hospital waste audit | Total knee joint replacements (surgical waste) | To investigate waste production associated with total knee replacements by performing a surgical waste audit to gauge the environmental impact of the procedure and generate strategies to improve waste management. | **Total surgeries:** 5 total knee joint replacements<br>**Mean waste weight per surgery, kg (%):**<br>• **Total waste: 13.3 (100)**<br>• Normal solid waste: 8.6 (64.5)<br>• Biohazard waste: 2.5 (19.2)<br>• Recyclable blue sterile wrap: 1.6 (12.1)<br>• Recyclable waste: 0.3 (2.2)<br>• Sharps: 0.3 (2.2)<br>**Data extrapolation:** Based on a volume of 1.6m$^3$ from the 5 surgeries, the landfill waste from all 47,429 total knee arthroplasties in Canada in 2008–2009 was estimated to be 407,889 kg by weight and 15,272 m$^3$ by volume. | Total knee arthroplasties produced substantial amounts of surgical waste. It was not maximally recycled, was improperly segregated and was associated with substantial surgical overage. |
| **Thiel et al. 2019** [90] | United States<br><br>**Setting:** Medical centre<br>**Timing:** Between May 2014 and July 2015 | Non-randomised, comparative analysis (including hospital waste audit) | Orthopaedic surgery (surgical waste) | To analyse the waste generation, material costs, and patient experience associated with wide awake hand surgery (WAHS) compared with surgery using traditional 'local & sedation' anaesthesia, while using a standard hand surgery custom pack or a minimal custom hand surgery pack. | **Total surgeries:** 178 small hand surgeries; carpal tunnel release (n = 80), trigger finger release (n = 39), cyst/mass excision (n = 32) and other (n = 27, includes de Quervain's release, Dupuytren's contracture treatments, nailbed or nerve repairs, or multiple procedures performed during a single surgical visit)<br><br>**Overall mean waste weight per surgery, kg (SD):**<br>Carpal tunnel release, trigger finger release and excision procedures: 2.4 (0.5)<br>'Other' procedures: 2.8 (0.4)<br><br>**Mean waste weight per surgery using 'standard pack' of disposable surgical supplies (n = 80), kg (SD)[26]:**<br>Carpal tunnel release, trigger finger release and excision procedures (n = 72): 2.6 (0.5)<br>'Other' procedures: 2.8 (0.4) | Implementing a 'minimal custom hand surgery pack' and wide-awake hand surgery (WAHS) together appear to halve surgical material costs for some commonly performed hand surgeries (carpal tunnel release, trigger finger release and excision of benign masses) and reduce mean surgical waste by 13% (0.3 kg) per case[28]. |

(*Continued*)

**Table 1.** (Continued)

| Author (year) | Country, setting and time of study | Study design | Topic | Aim/s and methods | Results | Conclusion |
|---|---|---|---|---|---|---|
| | | | | | **Mean waste weight per surgery using customised 'minimal pack' of disposable surgical supplies (n = 98), kg (SD)[27]:** Carpal tunnel release, trigger finger release and excision procedures (n = 72): 2.2 (0.5) 'Other' procedures: 2.8 (0.4) | |

Footnotes

**1:** Environmental impact data for five impact categories and one single score indicator were reported in graphical form only.

**2:** Nine patients from the combination anaesthesia group were analysed because one patient who received nitrous oxide was excluded.

**3:** Carbon emissions data did not include heating, ventilation, air conditioning or any surgical equipment.

**4:** Seven of the 305 patients lived more than 200 miles from the medical centre and were excluded from the analysis.

**5:** The central processing-related carbon footprint includes electricity usage for sterilisation and was calculated using data collected from the study institution.

**6:** The facility-related carbon footprint was calculated as the sum of the kg $CO_2$e produced when using operating room lights, anaesthesia equipment, endoscopy equipment, heating, cooling and the use of ventilation.

**7:** Waste-related carbon footprint was calculated as the sum of solid waste derived from positioning, prepping, draping, carpal-tunnel procedure, wound closure and wound dressing. The carbon footprint of solid waste was determined using the conversion factor of 0.199 kg $CO_2$e per kg.

**8:** eCTR requires more instrumentation than oCTR, resulting in fewer trays being sterilised per cycle and thus increasing its sterilisation energy requirements. eCTR also used more electricity compared with oCTR due to longer operating times.

**9:** This study also reported data for 481 speech therapy visits from a speech therapy unit, which are outside the scope of this review.

**10:** Based on the cost of a car being 0.26 kg $CO_2$e/km, derived from data by Leduc et al. (2010) [132].

**11:** Based on estimates by Lenzen et al. (1999) [133] that reported the cost of a car was 0.86 kg $CO_2$e/km.

**12:** This study did not perform carbon footprint calculations related to the number of plastic disposables used for each anaesthetic modality, the energy use for heating/cooling, ventilation, lighting, electricity for anaesthetic machines, surgical instruments, surgical implants, single-use items such as drapes and gloves, and intraoperative imaging.

**13:** Six general anaesthesia cases included the use of desflurane, which has a very high carbon footprint compared to other anaesthetic gases. Desflurane significantly skewed the mean total carbon footprint data for general anaesthesia.

**14:** The study also included plastic surgery procedures; breast reduction, breast augmentation, implant/expander removal, panniculectomy and abdominoplasty.

**15:** A blue wrap recycling program at Hahnemann campus, where collected blue wrap was sewn into charity items, diverted an additional 1.2 tonnes of waste from landfill over a 10-month period.

**16:** All surgeries included osteochondroplasty and labral repair.

**17:** Additional data were presented for the mean weight of other surgical procedures; ankle ORIF, humerus ORIF, clavicle ORIF, hip hemiarthroplasty and kyphoplasty, but the number of surgeries used to derive these data was not reported.

**18:** Fifty-two percent of the total surgical waste (110 grams) was related to surgical screws.

**19:** Waste weight data were converted from pounds to kilograms by multiplying figures by 0.454.

**20:** Contaminated waste items included surgical gloves, personal protective equipment, surgical drapes, tables, sponges, towels, tubing and surgical instruments.

**21:** Uncontaminated waste items included paper packaging, plastic packaging and blue polypropylene sterile wrap.

**22:** Hospital waste data for 223 non-musculoskeletal surgeries were also recorded for this audit.

**23:** Data are also available according to musculoskeletal surgery type.

**24:** This study also reported an additional waste audit of one total knee replacement, one laparoscopic procedure and one pelviscopic procedure, however, individual data could not be separated.

**25:** Domestic waste consisted of recyclable dry paper and card (47%), potentially recyclable plastic (47%) and non-recyclable wet paper or card or plastic (6%).

**26:** Weight of standard hand surgery custom pack was 2.04 kg.

**27:** Weight of customised minimal hand surgery pack was 1.62 kg.

**28:** There was no significant difference in mean waste weight between groups for "other" procedures (2-sample t test, P = 0.950).

Two LCA studies investigated the environmental impact of manufacturing a titanium implant for a knee [95] or hip [104] replacement. Both concluded that additive manufacturing of a prosthesis (building it one layer at a time) is more environmentally sustainable than creating complex geometric shapes using conventional methods (subtractive manufacturing or forging, milling, machining from a solid block of material until the final product is produced). One study reported that additive manufacturing of a titanium knee produced 68% less carbon emissions compared with conventional methods [95].

Two LCA studies investigated the carbon footprint of telehealth or telemedicine services versus usual care. One compared the carbon footprint of patient evaluations before and after implementing a model of care that included telehealth for patients undergoing elective spinal surgery [91], and the other compared the carbon footprint of telemedicine versus in-person consultations for hand surgery rehabilitation [97]. Both studies reported significant reductions in carbon emissions when telehealth or telemedicine was used.

Three LCA studies explored the carbon footprint of various hand surgeries [83–85]. One compared the carbon footprint of open to endoscopic carpal tunnel release surgery [85]. This study reported a significantly larger carbon footprint for endoscopic surgery due to higher energy requirements from sterilising surgical instruments and longer operating times. Another study quantified the carbon footprint of surgical waste from different types of hand surgeries and concluded it could be reduced by reducing the number of surgical items in the operating room and better sorting of waste for appropriate disposal [84]. The third study estimated the carbon footprint of three hand surgeries (carpal tunnel release, open reduction and internal fixation of distal radius fracture or primary flexor tendon repair) based upon the practices of 35 surgeons [83]. They found significant differences in operating room waste for the same surgery dependent upon the individual surgeon's practices.

One LCA from Germany compared the environmental impact of disposable versus reusable instrument sets for lumbar spine fusion surgery [103]. It found that disposable sets had 45 to 85% less environmental impact largely attributable to the high energy consumption of steam sterilisation for reusable sets.

One LCA was an engineering-based case study that included a multi-criteria decision-making approach to compare the environmental impact of three knee supports manufactured from different materials [105]. It concluded that these methods are useful to identify and optimise new eco-friendly products.

One LCA study quantified the average carbon dioxide equivalent ($CO_2e$) emissions of general, spinal and combination (general and spinal) anaesthesia used for knee replacements at a hospital in Melbourne, Australia using a 'cradle to grave' assessment [101]. This method measures the carbon footprint of a product from the resource extraction phase ('cradle') to its disposal ('grave'). As well as the anaesthesia, it included single-use items (e.g., plastics, glass, cotton etc.) and waste disposal. McGain et al. (2021) reported similar $CO_2e$ emissions for general, spinal, and combination anaesthesia when the parameters for the inhaled anaesthetic, including use of sevoflurane as the inhaled anaesthetic, were the same in those that received either general anaesthesia alone or a combination of general and spinal anaesthesia [101].

Their findings differed from another study performed in the US that found that the median $CO_2e$ emissions of general anaesthesia was significantly higher than spinal anaesthesia for single-level transforaminal lumbar interbody fusions (TLIF) [86]. This study performed a partial LCA using a 'cradle-to-gate' assessment, a method that only includes the carbon footprint of a product from the cradle to the moment that it is sold or received by the consumer ('gate'). Therefore, some large sources of $CO_2e$ emissions (e.g., single-use plastics, electricity for patient air warmer) were not included. Another point of difference was that the Australian study

calculated $CO_2e$ emissions based on an electricity mix derived from 75% brown coal which has double the $CO_2e$ emissions than electricity produced in the United States [101].

One Danish simulation study measured the energy consumption of differing types of ventilation (ventilation system fans and warming/cooling coils) in operating theatres during mock total hip replacements [102]. They reported that reducing fresh air influx for laminar airflow systems by 50% led to significantly lower energy consumption without resulting in an unacceptable increase in bacterial counts.

The remaining 12 research studies measured waste associated with orthopaedic surgery [87–90, 92–94, 96, 98–100, 106]. One study estimated that average dry weight waste, of which textiles (e.g. bandages, disposable sheets) accounted for over half, could be reduced from 5.7 to 4.5 kg per cemented hip replacement by switching to customised operating kits containing less consumable materials, packaging and products [98].

A further nine hospital waste audits quantified the weight of waste of 205 orthopaedic operations, predominantly joint replacements [88, 89, 93, 94, 99, 100, 106], but also hip arthroscopies [92], facet joint injections [100] and open reduction and internal fixation (ORIF) for malleolus ankle fracture [96]. Three waste audits reported the volume of surgical waste and extrapolated data to estimate annual landfill from knee replacement surgeries in Canada [94], as well as the potential reduction of waste or $CO_2$ emissions from recycling programs [99] and waste segregation [106]. Non-recyclable waste was the largest waste stream for most orthopaedic operations [88, 89, 92–94, 96, 99, 100, 106].

Most waste audits recommended strategies to reduce waste in orthopaedic surgery including implementing recycling programs [88, 92–94, 96, 99, 100, 106], segregating waste [88, 92–94, 96, 99, 100, 106], educating hospital staff to correctly dispose of recyclable waste [88, 93, 100], documenting 'green outcomes' from surgical procedures to encourage green health care practices [92], commencing surplus recovery programs [92, 94], reducing excessive packaging of surgical materials [94, 96, 98, 106], moving to reusable surgical linens [94], providing surgeons with a selection of operating kits that can be re-sterilised between procedures [96], and adopting new procurement routines [98, 100].

One study found that combining a 'minimal custom' surgery pack with local anaesthesia rather than a standard surgery pack with sedation and local anaesthesia reduced average surgical waste by 13% for minor hand surgery [90]. The final study redesigned the operating set to include 23 rather than 35 instruments for hand surgery and implemented a waste recycling program that resulted in a 20 to 51% increase in monthly recycling rates across three hospital sites [87].

**Editorials.** Twenty-one editorial papers described a need to reduce environmental impact of orthopaedic surgery (n = 9) and focussed on disciplines responsible for managing musculoskeletal conditions (n = 12). Of those relating to orthopaedic surgery; three discussed a need for orthopaedic surgery to adopt sustainable practices [121, 125, 126]; two discussed strategies for reducing the environmental impact of hand surgery [120, 122]; one focused on the benefits of regional anaesthesia in place of inhaled volatile anaesthetic gases [124]; one discussed the reuse of undamaged surgical screws or prostheses opened but not used during surgery [123]; one discussed recycling of metal implants posthumously [117]; and one reported the total weight of waste from 1,099 unspecified hand surgeries, but no methods were reported [127]. Of those discipline-specific editorials, nine discussed the environmental impact of physiotherapy and the role of the profession in reducing it [109–111, 113–116, 118, 119], two discussed how podiatrists can engage with the community to drive sustainable practice [107, 108], and one outlined strategies for occupational therapists to approach climate change [112].

**Narrative reviews.** Two narrative reviews summarised environmentally sustainable changes that can be implemented for hand surgery [128, 131]. One summarised 'Lean and

Green' initiatives that aim to reduce waste-energy consumption, improve sterilisation techniques and reprocess single-use devices [128], and the other summarised changes to reduce the carbon footprint of hand surgery using a 'Reduce, Reuse, Recycle, Research, Rethink and Culture' framework [131]. Both reviews reported financial benefits from implementing environmentally sustainable hand surgery practices. There were four additional papers relating to these publications [87, 90, 94, 127] (see Tables 1 and S3).

One narrative review on environmental sustainability in orthopaedic surgery, that identified all seven relevant studies that we included, highlighted a need for high-quality research on best practices for orthopaedic surgery to reduce its carbon footprint [129] (see Table 1). The remaining narrative review explored bioresorbable orthopaedic implants as a sustainable alternative to traditional permanent implants for some orthopaedic surgeries [130].

## Discussion

Our scoping review identified 49 publications focused on the environmental impacts of health care for musculoskeletal conditions. Most papers were published within the last three years and almost half were editorials, likely reflecting an increasing interest in the topic. Almost three-quarters were related to orthopaedic surgery which is consistent within other health fields that have recognised surgery as a large contributor of greenhouse gas emissions [134–136]. Of the 24 included original research studies less than half directly measured the environmental impact relating to climate change for any aspect of musculoskeletal health care and none quantified the carbon footprint of well-recognised contributors of greenhouse gas emissions such as hospital stays, pharmaceuticals and imaging [4, 55].

Our review identified some promising strategies for reducing the environmental impact of musculoskeletal health care including use of additive rather than subtractive manufacturing of orthopaedic components, greater use of telehealth, and reducing fresh air influx for laminar airflow systems in operating theatres, that warrant further investigation. The finding that open carpal tunnel release has a lower carbon footprint compared to endoscopic release, which may be preferred by the patient [137], indicates a need to consider these competing priorities. Similarly, while many studies identified ways of reducing waste in orthopaedic surgery including greater use of reusable instruments, the finding from one study that reusable instrument sets had a greater carbon footprint in comparison to disposable sets indicates that evidence of environmental benefit is required before introducing changes to practice.

To better understand the environmental impact of health care for musculoskeletal conditions there is a need to identify and quantify the impact of care in terms of a carbon footprint, and implement standardised and valid metrics for routine collection across multiple institutions and government bodies [138, 139]. Collecting comparable carbon metrics associated with the delivery of musculoskeletal care such as $CO_2e$ emissions via life cycle assessment or the development of new carbon intensity metrics would facilitate accurate benchmarking, monitoring and transparent reporting of data that can be used to identify high emitters of greenhouse gases for targeting efforts to reduce them [138, 139]. The methods for collecting these metrics are complex and, as exemplified by the different results in comparing general to spinal anaesthesia across countries and by use of different LCA methods (cradle to gate or to grave metrics), specialised expertise is needed to be able to explain such differences.

Nine of the original research studies included in this review were waste audits that provided some information regarding the weight, volume and type of hospital waste associated with orthopaedic surgeries. However, the estimates had low precision and poor generalisability as they were based on a small number of surgical operations ranging from one to 55. While larger studies performed across multiple hospital sites would provide more representative samples of

the waste produced from orthopaedic surgeries, the UK National Health Service (NHS) estimates that waste produced across the NHS healthcare system accounts for only 3% of the total carbon footprint of health care [7]. Future research on the environmental sustainability of orthopaedic surgery may therefore have greater impact if directed towards larger contributors of greenhouse gas emissions. Additionally, there may be opportunity for existing waste audit data to be quantified as $CO_2$e emissions estimates using retrospective life cycle assessment methods, although this process requires a high level of expertise and is resource intensive [139].

Environmentally sustainable health care is needed across all health systems to minimise the direct and indirect harms it may be causing to our planet and its population [140]. In addition to collecting meaningful data using standardised carbon metrics, a framework by MacNeill et al. (2021) proposes three principles for achieving health system sustainability that can be directly applied to musculoskeletal care [141]. The first principle involves reducing the demand for health services. While this has grown as a consequence of ageing populations and population growth, public health policies are needed that prioritise disease prevention which will have additional benefits beyond musculoskeletal health. The second principle is to better match the supply and demand of health care and health support services across populations and settings, while the third is to reduce greenhouse gas emissions from the delivery of health care. The latter could be achieved by de-implementation of low value care, particularly targeting low-value tests and treatments with large carbon footprints, as well as expanding low carbon services such as telehealth across health systems. Many of the publications included in this review align with this third principle, although more carbon metrics are needed to further determine the largest contributors of greenhouse gas emissions within musculoskeletal health care.

The main strength of this review is that we used scoping review methodology to identify a broad range of studies and editorials across multiple disciplines. We also developed a comprehensive environment-themed search strategy through discussion with environmental scientists and after examining systematic reviews that had explored environmental sustainability for health care in other fields [2, 142]. We did this because we could not identify validated search strategies published for 'environmental health' or 'environmental impact'.

A limitation to our database search is that we used the search strategy for musculoskeletal conditions used by Cochrane Musculoskeletal [143, 144], but this did not include broad anatomical terms (e.g. hand, wrist, elbow, shoulder etc.). To overcome this, we performed comprehensive Google and Google Scholar searches using anatomical, surgical, telehealth and environment themed keywords and also hand searched the reference lists of included publications to identify relevant publications and grey literature articles not published or indexed in biomedical databases. Our search identified narrative reviews that included 11 of our included original research studies and no additional relevant papers also minimising the likelihood of missing papers that would have appreciably altered our conclusions.

## Conclusion

Despite an established link between health care and greenhouse gas emissions we found limited empirical data estimating the impact of musculoskeletal health care on the environment. Most of the studies we identified quantified the carbon footprint of aspects of orthopaedic surgery, particularly surgical waste, but there were limited data for other aspects of care such as imaging, pharmaceuticals and allied health care. Further data are needed to determine whether actions to lower the carbon footprint of musculoskeletal health care should be a priority and to identify those aspects of care that should be prioritised.

## Supporting information

**S1 Data. PRISMA-ScR checklist.**
(DOCX)

**S2 Data. Search strategies.**
(DOCX)

**S1 Table. Excluded studies.**
(DOCX)

**S2 Table. Studies awaiting assessment.**
(DOCX)

**S3 Table. Focus and conclusions of editorials.**
(DOCX)

**S4 Table. Focus and conclusions of literature reviews.**
(DOCX)

## Acknowledgments

We would like to thank Diane Horrigan for her assistance in developing the database search strategies. We would also like to thank Zoe Rammelkamp for providing raw waste weight data of surgeries reported in the hospital waste audit by Rammelkamp et al. (2021) [89].

## Author Contributions

**Conceptualization:** Rachelle Buchbinder.

**Data curation:** Bayden J. McKenzie, Romi Haas, Giovanni E. Ferreira, Rachelle Buchbinder.

**Formal analysis:** Bayden J. McKenzie.

**Investigation:** Bayden J. McKenzie, Romi Haas, Giovanni E. Ferreira, Chris G. Maher, Rachelle Buchbinder.

**Methodology:** Bayden J. McKenzie, Rachelle Buchbinder.

**Project administration:** Bayden J. McKenzie.

**Supervision:** Romi Haas, Giovanni E. Ferreira, Chris G. Maher, Rachelle Buchbinder.

**Writing – original draft:** Bayden J. McKenzie.

**Writing – review & editing:** Romi Haas, Giovanni E. Ferreira, Chris G. Maher, Rachelle Buchbinder.

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
