## [Decision Letter · Decision Letter 0]

29 Mar 2022

PONE-D-22-02641The environmental impact of health care for musculoskeletal conditions: a scoping reviewPLOS ONE

Dear Dr. McKenzie,

Thank you for submitting your manuscript to PLOS ONE. After careful consideration, we feel that it has merit but does not fully meet PLOS ONE’s publication criteria as it currently stands. Therefore, we invite you to submit a revised version of the manuscript that addresses the points raised during the review process. Thank you for your time and effort on this manuscript. Please review the following comments along with those of the reviewers if you opt to re-submit for a major revision. The following comments are required to be addressed for consideration of publication on resubmission. Greatly appreciate your interest and effort on sustainable healthcare.

1) Among the reviewers there is a concern regarding your search strategy. Please expand your search methodology to look for manuscripts which may be related to musculoskeletal pathology and sustainability but not accessible with those search terms. As you identify manuscripts that you did not initially capture, you may need to include additional search terms / broaden your search in your scoping review.

— I note that you write in your limitations that you do not think you missed any manuscripts but please consider reviewer 1’s comment and references.

2) Please include a section related to sustainability knowledge that does not directly pertain to musculoskeletal conditions but can be applied to musculoskeletal conditions and treatment.

3) Please make sure the McGain et al. TKA LCA research is appropriately represented as there is much nuance in that article that needs to be understood and communicated.

4) Please expound upon future steps that are necessary to provide more sustainable musculoskeletal care and be specific. In the situation where a dearth of information is present, the most valuable part of this manuscript may be directing researchers where to look and what would be considered valuable.

5) Figure 1 is not legible in the PDF form of the submission

We look forward to receiving your revised manuscript.

Kind regards,

Matthew John Meyer, M.D.

Academic Editor

PLOS ONE

Journal Requirements:

(BM is supported by a PhD scholarship from the Chiropractic Australia Research Foundation and a top-up scholarship from the National Health and Medical Research Council (NHMRC) Australia & New Zealand Low Back Pain Research Network Centre of Research Excellence (ANZBACK CRE) (1171459). GF is supported by an NHMRC Emerging Leadership fellowship (APP2009808). CM and RB are supported by NHMRC Leadership Fellowships. The funders had no role in study design, data collection and analysis, decision to publish, or preparation of the manuscript.

Funder websites: https://chiropracticaustralia.org.au/research-foundation/, https://anzback.org/, https://www.nhmrc.gov.au/)

Reviewers' comments:

Reviewer's Responses to Questions

**Comments to the Author**

1. Is the manuscript technically sound, and do the data support the conclusions?

Reviewer #1: Partly

Reviewer #2: Yes

2. Has the statistical analysis been performed appropriately and rigorously? 

Reviewer #1: Yes

Reviewer #2: Yes

3. Have the authors made all data underlying the findings in their manuscript fully available?

Reviewer #1: Yes

Reviewer #2: Yes

4. Is the manuscript presented in an intelligible fashion and written in standard English?

Reviewer #1: Yes

Reviewer #2: Yes

5. Review Comments to the Author

Reviewer #1: Thank you for preparing your thoughts and analysis for peer review. Your paper covers an area that is needed, and has little presence in the literature to date, so appreciate your review of this topic. In performing a brief search on this topic, I have found articles (below) that may deserve consideration in your review. The hallmark of a narrative review is that it is sufficiently comprehensive as to not leave notable papers out of the analysis in the date range selected, and presents the breadth of literature intended. This is not an insistence that you include these references below, but a mere example of papers that I am seeing that appear relevant, but will need further looking into on your part to see if they merit inclusion.

Additionally, it may be worth noting that the most significant article you highlight (total knee arthroplasty comparison of regional vs. general anesthesia, F McGain…first paper in Table 1 of your paper) I know this paper well, and when citing this article it is essential to recognize that regional and general anesthesia are only similar in carbon footprint (as the authors conclude) after implementing significant sustainability strategies, which in this study was to omit all anesthetics where other gases were used at their center (they used the lowest impact gas only, which is Sevoflurane, and at an average low fresh gas flow, and threw out all the others in the analysis…such as Desflurane, Nitrous Oxide, etc.). Otherwise, general anesthesia would be considered much higher on carbon impact in many other settings. I don’t know that you need to get into this level of detail in your review, but should be aware of this as you choose to refer to this article and understand how to cite it. In addition, please double check your referencing F McGain’s paper in saying that “ The largest contributors to the (line 143) carbon footprint across all groups were single-use equipment such as plastics..”….as according to Table 2 in that paper you cited, the carbon footprint for general anesthesia was greatest (35%) for anesthetic gases, and second largest was single-use disposables (closer to 28% if you add up the items in that table). This is a minor point, perhaps, but just wanted you to be aware for accuracy.

1) This review below cites several publications that may be relevant to your search criteria:

Alisina Shahi, et al. AAOS Now, April 2021. What Is Orthopaedic Surgery’s Environmental Impact?

https://www.researchgate.net/profile/Haley-Tornberg/publication/350771203_What_Is_Orthopaedic_Surgery%27s_Environmental_Impact/links/6070a14a4585150fe997fb76/What-Is-Orthopaedic-Surgerys-Environmental-Impact.pdf?origin=publication_detail

included in that bibliography is this article that may be worth considering:

2) Pavlou P, Gardiner J, Pili D, et al: The environmental impact of large joint arthroplasty. Orthopaedic Proceedings 2010;92-B(SUPP_IV):498.

https://online.boneandjoint.org.uk/doi/abs/10.1302/0301-620X.92BSUPP_IV.0920498

Overall, I think you’ve done a service to the profession by working on this review, and feel you have done a credible job with it- just wanted to pass off some items to consider before publishing.

Reviewer #2: Thank you for preparing this paper - a scoping review of the environmental impacto musculoskeletal health care.

At a big picture level, this paper on one hand is about an important topic - the environmental contribution of health care, but on the other - doesn't add any new information, and instead just finds that there isn't really any useful data out there. Not the fault of the authors, but as a reader, it doesn't really satisfy. I think perhaps it is a consequence of the search strategy. ie people might not carve out musculoskeletal health in this way, but more hospital, outpatient care, radiology etc.

I think a more useful paper would be to try and create a construct of environmental impacts. ie outpatient care visits, radiology , in hospital, operations etc.

It would be more interesting to have a scoping review that outlines the contributions of musculoskeletal health care ( although separating the components of healthcare is only sort of interesting in identifying targets for change) to the environment. ie

- there is a lot written about green house gases and operations - many musculoskeletal procedures end up needing operations. However, this literature is not specific about musculoskeletal conditions as it is the OT / operation.

- there is literature out there about virtual visits. Especially follow-up visits could be switched to this mode of delivery.

Table 1 sort of provides this framework, and might be a better way to structure the concept amongst the paucity of data.

6. PLOS authors have the option to publish the peer review history of their article (what does this mean?). If published, this will include your full peer review and any attached files.

Reviewer #1: No

Reviewer #2: No

---

## [Author Response · Author response to Decision Letter 0]

14 Aug 2022

Specific responses to reviewer and editor comments can be found in the uploaded Word document titled 'Response to reviewers'.

---

## [Decision Letter · Decision Letter 1]

4 Sep 2022

PONE-D-22-02641R1The environmental impact of health care for musculoskeletal conditions: a scoping reviewPLOS ONE

Dear Dr. McKenzie,

Thank you for submitting your manuscript to PLOS ONE. After careful consideration, we feel that it has merit but does not fully meet PLOS ONE’s publication criteria as it currently stands. Therefore, we invite you to submit a revised version of the manuscript that addresses the points raised during the review process.

 Thank you very much for the hard-work and time you invested to improve this manuscript. You found a number of interesting studies that are related to the musculoskeletal system and sustainability. My goal for the next draft is to take the additional information that is well-presented in Table 1 and incorporate it into the text of the manuscript (specifically results / discussion / conclusion). Please synthesize the data you have found for the reader. Additionally, the sentence structure is often complex and long with multiple clauses including a variety of punctuation (ex. parentheticals) and sometimes even thoughts--simplify to information that is educational for the reader. Find below some additional comments to help with the next draft.

We look forward to receiving your revised manuscript.

Kind regards,

Matthew John Meyer, M.D.

Academic Editor

PLOS ONE

Journal Requirements:

Additional Editor Comments:

TABLE 1

Wang et al.

Total surgeries: 100 by a single surgeon (50 s general and 50 spinal anaesthesia)

“S” after 50 is a typo

Median total carbon footprint, grams CO2e Spinal anaesthesia12: 70

70g of CO2 seems to be too little--is this just considering the spinal needle and the local anesthetic? What about the drapes, O2 nasal cannula, O2 gas…? Especially when compared to the McGain Anesthesiology paper (and these two should be discussed together), there is a contradiction in findings.

Similarly 231-234 …one retrospective study from the United States reported that the median total carbon footprint of general anaesthesia was significantly higher when compared to spinal anaesthesia (4,725 grams CO2e versus 70 grams CO2e) for a single-level transforaminal lumbar interbody fusion [53]...

RESULTS

In all results subsections, but specifically “original research studies” please collate and present consistent themes from the conclusions of the manuscripts. The results section reads like a bibliography (specifically sentence two of paragraph one (214-226) and presents the topic of the article found, but very little about the lessons of that article. You have done the identification of the important points in “results” and “conclusions” columns in Table 1. Please put these conclusions together for the reader in results to assist in hypothesis-generation for future research.

As an example, additive manufacturing is mentioned at least twice as being less carbon intensive. This is a new-ish technology which has multiple applications in the medical world--I was not aware how much more sustainable it is considered--there is value in sharing this.

DISCUSSION

333-344 are two very complicated sentences requiring simplification for comprehension.

CONCLUSION

The following sentence isn't quite right because it implies we should expect clarity in other segments of health care whereas the assessments and connections are frustratingly basic in health care and other industries--linking GHG emissions to any complex process is very challenging.

“Despite an established link between health care and greenhouse gas emissions, the carbon footprint of health care for musculoskeletal conditions is unknown.”

You have now found some manuscripts with the additional research--what did you find, and what do you want to be the specific next steps?

Reviewers' comments:

Reviewer's Responses to Questions

**Comments to the Author**

1. If the authors have adequately addressed your comments raised in a previous round of review and you feel that this manuscript is now acceptable for publication, you may indicate that here to bypass the “Comments to the Author” section, enter your conflict of interest statement in the “Confidential to Editor” section, and submit your "Accept" recommendation.

Reviewer #1: All comments have been addressed

Reviewer #2: All comments have been addressed

2. Is the manuscript technically sound, and do the data support the conclusions?

Reviewer #1: Yes

Reviewer #2: Yes

3. Has the statistical analysis been performed appropriately and rigorously? 

Reviewer #1: Yes

Reviewer #2: N/A

4. Have the authors made all data underlying the findings in their manuscript fully available?

Reviewer #1: Yes

Reviewer #2: Yes

5. Is the manuscript presented in an intelligible fashion and written in standard English?

Reviewer #1: Yes

Reviewer #2: Yes

6. Review Comments to the Author

Reviewer #1: Initial concerns have been adequately addressed- appreciate your work on this and I am recommending publication. Thank you for your submission.

Reviewer #2: Unfortunately I couldn't find the response to reviewers page amongst the uploaded documents, but you have largely addressed my concerns.

7. PLOS authors have the option to publish the peer review history of their article (what does this mean?). If published, this will include your full peer review and any attached files.

Reviewer #1: **Yes: **Samuel J Smith, MD, MPH

Reviewer #2: No

---

## [Author Response · Author response to Decision Letter 1]

4 Oct 2022

Reviewer 1: Thank you for your feedback. We are pleased that your initial concerns were adequately addressed and that you are recommending publication.

Reviewer 2: Thank you for your comments. We are pleased that your initial concerns were largely addressed.

Editor: Thank you for your further feedback of our paper titled: The environmental impact of health care for musculoskeletal conditions: a scoping review, PONE-D-22-02641. We have uploaded a revised manuscript with and without track changes that includes all of your suggestions relating to the results, discussion and conclusion sections. Your feedback was very helpful and we believe has improved our review.

Point-by-point responses to your feedback are outlined below.

TABLE 1

Wang et al.

Total surgeries: 100 by a single surgeon (50 s general and 50 spinal anaesthesia)

“S” after 50 is a typo

Authors' response: Thank you for picking up this typo. It has been corrected in the manuscript.

Median total carbon footprint, grams CO2e Spinal anaesthesia12: 70

70g of CO2 seems to be too little--is this just considering the spinal needle and the local anesthetic? What about the drapes, O2 nasal cannula, O2 gas…? Especially when compared to the McGain Anesthesiology paper (and these two should be discussed together), there is a contradiction in findings.

Similarly 231-234 …one retrospective study from the United States reported that the median total carbon footprint of general anaesthesia was significantly higher when compared to spinal anaesthesia (4,725 grams CO2e versus 70 grams CO2e) for a single-level transforaminal lumbar interbody fusion [53].

Authors' response: Thank you for highlighting the apparent inconsistencies of results between the McGain et al. (2021) and Wang et al. (2022) studies. 

We agree that the 70 grams of CO2e emissions for spinal anaesthesia during single-level TLIFs as reported by Wang et al. is small but is likely explained by the study design which was a partial LCA known as a “cradle-to-gate” assessment. This method includes the carbon footprint of a product from the resource extraction phase (“cradle”) to the moment it is sold or received by the consumer (“gate”). Therefore, the environmental impact of anaesthetic machines, cleaning equipment, compression of liquid oxygen, plastic disposable items or waste disposal was not considered.

In contrast, the higher values reported by McGain et al. is because their LCA was a “cradle-to-grave” assessment. This includes the carbon footprint associated with using the product and its disposal (“grave”). 

Another reason for the discrepancy in CO2e emissions is likely explained by different sources of electricity used in Australia and the United States. According to McGain et al., electricity generated in Victoria, Australia generates twice the CO2e emissions when compared to the US).

We have updated the results section to include an explanation of the likely reasons for the differing results (please see lines 304 to 313).

RESULTS

In all results subsections, but specifically “original research studies” please collate and present consistent themes from the conclusions of the manuscripts. The results section reads like a bibliography (specifically sentence two of paragraph one (214-226) and presents the topic of the article found, but very little about the lessons of that article. You have done the identification of the important points in “results” and “conclusions” columns in Table 1. Please put these conclusions together for the reader in results to assist in hypothesis-generation for future research.

As an example, additive manufacturing is mentioned at least twice as being less carbon intensive. This is a new-ish technology which has multiple applications in the medical world--I was not aware how much more sustainable it is considered--there is value in sharing this.

Authors' response: We had not wanted to overburden the reader by duplicating the results and conclusions columns of Table 1 in the text. However, we agree that reporting more details from the original studies is worthwhile and we have now included this information in the text of the manuscript for all LCA and carbon footprint studies methods (see lines 225 to 313). We have also updated the results for the two LCAs that reported a significantly reduced environmental impact of additive manufacturing compared to conventional methods (see lines 225 to 231).

We did not expand the editorial and narrative review subsections of the results as their details are described in Tables S3 and S4 but we can do so if the editor wishes us to.

DISCUSSION

333-344 are two very complicated sentences requiring simplification for comprehension.

Authors' response: We agree and have now revised and simplified these sentences (see lines 451 to 465).

CONCLUSION

The following sentence isn't quite right because it implies we should expect clarity in other segments of health care whereas the assessments and connections are frustratingly basic in health care and other industries--linking GHG emissions to any complex process is very challenging.

“Despite an established link between health care and greenhouse gas emissions, the carbon footprint of health care for musculoskeletal conditions is unknown.”

Authors' response: We agree that this is a complex process. We have now revised the conclusion to better match the conclusion in the abstract (see lines 502 to 509). We have also added text to the third paragraph of the discussion indicating that measuring the environmental impact of health care is complex and requires specialised expertise (see lines 428 to 431).

You have now found some manuscripts with the additional research--what did you find, and what do you want to be the specific next steps?

We have added a paragraph in the discussion to summarise what we found overall and promising avenues for further research (see lines 406 to 416).

---

## [Editor Report · Decision Letter 2]

12 Oct 2022

The environmental impact of health care for musculoskeletal conditions: a scoping review

PONE-D-22-02641R2

Dear Dr. McKenzie,

We’re pleased to inform you that your manuscript has been judged scientifically suitable for publication and will be formally accepted for publication once it meets all outstanding technical requirements.

Kind regards,

Matthew John Meyer, M.D.

Academic Editor

PLOS ONE

Additional Editor Comments (optional):

Thank you for all the additional effort. I think the manuscript reads much better. I truly appreciate your hard work in this area and hope you continue to expand upon the topic. Few thoughts below:

95-96: We did not impose any date or language restrictions  regarding "language" the search terms in S-Data2 are in English--did you include any non-English manuscripts? If accurate, please keep "language". If not, eliminate. The search undeniably returned international results but were they multi-lingual?

269-270: "sevoflurane was used as the inhaled anaesthetic for general and combination approaches and an average low fresh gas flow was used [68]" I know what you are trying to say regarding the McGain paper and "low-flow" but this will not be clear to a reader who is unfamiliar with the manuscript. Perhaps be more specific about the parameters of flow (mL/min) used on average or re-work the syntax of the last clause.

I appreciate the detail in S-Table2 allowing those really interested to pursue data that is about to come out. S-Table3 has a typo "carviovascular" in the Maric et al. citation fourth column.
---

## [Editor Report · Acceptance letter]

15 Nov 2022

PONE-D-22-02641R2 

The environmental impact of health care for musculoskeletal conditions: a scoping review 

Dear Dr. McKenzie:

I'm pleased to inform you that your manuscript has been deemed suitable for publication in PLOS ONE. Congratulations! Your manuscript is now with our production department. 

Kind regards, 

on behalf of

Dr. Matthew John Meyer 

Academic Editor

PLOS ONE